# Effect of an Innovative Biofeedback Insole on Patient Rehabilitation after Total Knee Arthroplasty

Gianluca Castellarin [1], Michele Merlini [1,*], Giulia Bettinelli [2], Raffaella Riso [1], Edoardo Bori [3] and Bernardo Innocenti [3]

1   Department of Orthopaedic Surgery II, Suzzara Hospital, Via G. Cantore 14, 46029 Suzzara, Italy;
    gianlucacastellarin@gmail.com (G.C.); riso.raffaella@gmail.com (R.R.)
2   Vita-Salute San Raffaele University HSR, Via Olgettina 58, 20132 Milan, Italy; bettinelli.giulia@hsr.it
3   Bio Electro and Mechanical Systems (BEAMS) Department, Université Libre de Bruxelles,
    1050 Brussels, Belgium; edoardo.bori@ulb.be (E.B.); bernardo.innocenti@ulb.be (B.I.)
*   Correspondence: merlini_michele@libero.it

**Abstract:** Partial weight bearing is fundamental to rehabilitation in the early stages following lower limb surgery. However, it remains debated as to how to properly achieve partial weight bearing while avoiding complications from excessive or premature load. Of the devices currently on the market, instrumented insoles coupled with force-sensitive resistors (FSRs) are among the best options in today's clinical practice. Still, although several of these systems have been developed in the last few years, only some have been validated, leaving insufficient information on their application in rehabilitation after total knee replacement (TKR). To address this research gap, we evaluated the performance of an innovative biofeedback insole system featuring an extremely low response time for real-time force feedback. We randomly recruited 30 patients who underwent total knee arthroplasty. All patients used the new programmable insole for partial weight bearing per post-operative rehabilitation protocol. Our results confirm their inability to perform a correct gait with low partial weight bearing (<30–50% of their bodyweight). Partial weight bearing with a correct gait in the post-operative period is not obtainable without a measuring system. This new biofeedback insole is thus one of the most indicated and can improve rehabilitation compliance, therefore allowing continual patient monitoring for faster discharge and fast-track rehabilitation.

**Keywords:** biofeedback; knee arthroplasty; partial weight bearing; fast-track rehabilitation





## 1. Introduction

Proper observation of patients' weight bearing is difficult to attain after lower limb surgery as it is challenging to control their compliance with the recommended rehabilitation protocol in a domestic and active functional environment. Moreover, it is well known that after major lower limb surgery, such as total hip or knee arthroplasty, the load on the affected limb is at least always >30% of the patient's bodyweight, thus leading to possible loosening or failure of the prostheses [1–6].

Any lower limb surgery (from the femur to the foot), either elective or traumatologic, is followed by a partial weight bearing (PWB) period on the treated limb, which has multiple aims widely described and analyzed in the literature [4]. Most notably, PWB permits correct osteo-integration; reduces implant failure, aseptic mobilization, and implant breakage; and promotes wound healing and pain control. At the same time, the PWB period prevents prolonged discharge (non–weight bearing) and related complications on the operated limb [2–8].

In addition to traditional monitoring methods, new feedback methods have been developed on the basis of scales or tactile/verbal feedback from physical therapists, although they remain extremely operator-dependent and are scarcely reproducible. Initially, these methods used plates capable of measuring the patient's weight bearing force: unfortunately,

those systems were cumbersome and thus used only in laboratories for research purposes. Recently, with the development of more advanced technologies and force-sensitive resistors (FSRs), it has been possible to design simpler tools that can even be worn as footwear soles and linked to wearable control units [9–12].

The proper use of FSR in rehabilitation practice following total knee replacement (TKR) has not been comprehensively evaluated yet. Considering the exponential increase in the number of TKRs performed worldwide and the lowering of patient age to receive this surgery due to both improved surgical techniques and prosthesis designs, increasingly more patients have high functional expectations or are still active workers. For these patients in particular, it is desirable to resume activities and return to their domestic environment as quickly as possible [13]. To adequately respond to these new societal demands, a multidisciplinary approach (different anesthesia modalities/administrations, wound dressing improvements, better post-operative analgesia, better surgical techniques) led to the development of "fast-track" protocols for patient management after TKR, reducing, in some cases, the post-operative hospital stay from an average of 2 weeks to 1 week (with 75% of patients discharged within 3 days post-op) [14]. However, early patient discharge poses challenges in controlling compliance with recommended rehabilitation protocols, including PWB [13].

The aim of this study was to verify if the use of an insole able to provide feedback on the amount of load exerted on the operated limb could improve patient compliance and cooperation during rehabilitation and improve the quality of their recovery.

The hypothesis was that such a device could have beneficial effects both on the patient rehabilitation process and on the ability of the physiotherapist to implement the required recovery process.

## 2. Materials and Methods

This study was conducted according to the guidelines of the Declaration of Helsinki (7th revision; 64th WMA General Assembly, Brazil, October 2013) and in observance of existing Italian legislation on privacy and sensitive data processing. Prior informed consent was obtained from all subjects involved in the study.

From November 2019 to February 2020, we enrolled 30 patients diagnosed with gonarthrosis (stages C and D according to the International Knee Documentation Committee, IKDC) and who were scheduled to receive a total joint replacement. It was not possible to blindly collect the results with a control arm due to the obvious presence of the insole and collection device. However, in order for any possible confounding factors (i.e., bias) to be reduced, the patients were operated on by the same surgeon (G.C.) using the same surgical technique, which consists of an extra medullary guide (EMAS) already described in previously published papers [15,16], the same approach, and the same prosthesis (Genus mobile bearing knee with LS insert, Adler Ortho SpA, Cormano, Italy) [15–18]. In all cases the Posterior Cruciate Ligament (PCL) was spared. All prosthetic components were cemented, and the patella was never resurfaced.

Of the 30 enrolled patients (20 females,10 males), the mean age was 69.5 years (min.: 60, max.: 83). Eight patients had their left knee replaced, and the remaining 22 had their right knee replaced; the mean bodyweight was 69.28 ± 12.9 kg. Patient selection was performed according to the following inclusion criteria: age < 85 years, no neurological or orthopedic pathologies other than knee arthrosis, and no comorbidities or other issues that could lead to low compliance with the insole and the rehabilitation protocol. A total of 23 patients (77% of the cohort) had pre-operative varus deformity > 5°, 3 patients (10% of the cohort) had a varus deformity between 3° and 5°, and 4 patients (13% of the cohort) had a valgus deformity between 3° and 5°. Only one patient was excluded halfway through the study due to the development of a severe neurological deficit in active hip flexion; this cruralgia relapse was secondary to a pre-existing lumbar disc pathology and was non-responsive to pharmacological therapy. Another patient had minor problems using the insole because of

an ankle sprain that occurred in the immediate pre-operative period, which significatively reduced the patient's compliance.

The feedback system was the Blu Insole (FGP srl, Dossobuono 37062, VR, Italy); Figure 1). The device is made of thin (<5 mm), flexible polyester available in different shoe sizes (European sizes 35–48) and two sides (left and right) that can be easily cut and shaped to fit each patient's foot and any kind of shoe. Compared to previous commercial designs [9,10] that enable the use of a maximum 13 FSRs, the Blu Insole allows the use of 214 FSRs, improving the overall quality of the measurement. FSRs are designed to measure load at a frequency of 10 Hz and response time of about 0.2 s. This extremely low response time makes it is possible to train patients to autonomously correct their weight bearing deficits or excesses. The insole is connected to a capture device using a flexible connector; the capture device is fixed to the same patient's ankle with an adjustable ankle band (Figures 1 and 2).

The device is CE mark, and it has been developed and produced according to the EU standards 93/42/CEE.

Using an appropriate software provided by the manufacturer and connecting the insole to the operator PC by means of an USB cable, one can program the unit to provide patients audible and tactile (vibration) feedback on two different thresholds: one when they succeed in gaining the minimum requested load, and one when they exceed the maximum weight allowed. In practical terms, the operator enters the patient body weight and then sets up the maximum percentage of it allowed before the patient will feel the vibrating feedback. Therefore, it is possible to train patients to reach a "permitted load" range that can be defined in advance by medical or physical therapy staff according to the patients' needs. The system also has a rechargeable battery with a half-life of about 40 h, roughly corresponding to 5 days of use during the immediate TKR post-operative period. The battery can be recharged by means of as USB cable similar to the ones used for mobile phones.

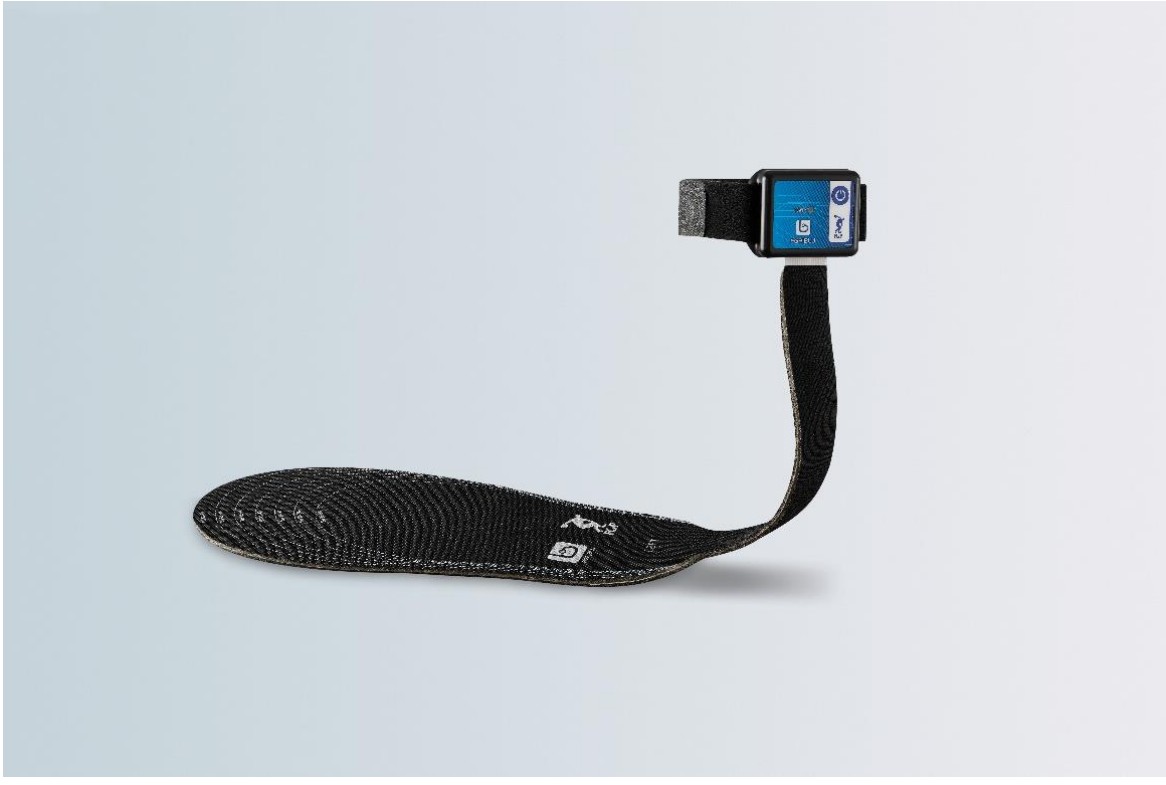

**Figure 1.** The insole. The device is composed of the insole containing the weight sensors, which is connected with a band to the electronic bio-feedback display. The display is then attached to the patient leg with a Velcro strap.

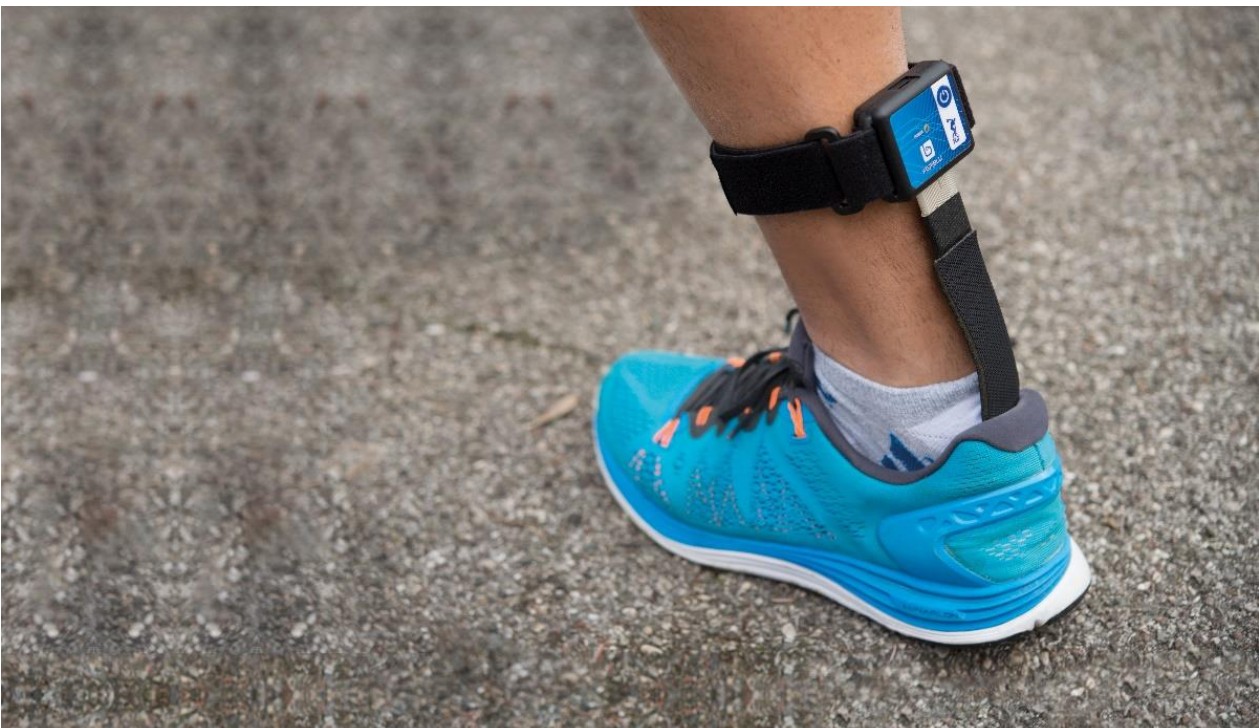

**Figure 2.** How the insole and the display should be worn by the patient.

The enrolled patients were evaluated at time 0 (pre-surgery, upon hospital admission) with the insole to confirm their ability to reach complete weight bearing. They were also instructed on its functioning, modeling capacity, and calibration. On the fourth day post-op, the patients were transferred to our clinic's rehabilitation ward and evaluated again by the physical therapist and physiatrist team. Evaluations were carried out at 5, 10, 15, and 30 days post-op using the Numerical Rating Scale (NRS, from 0 to 10) to assess pain; the Knee Score System (KSS, updated 2011), including both the comprehensive objective standard evaluation and the functional component, each with a score of 0–100 [19]; and the Western Ontario McMaster Universities Index Osteoarthritis (WOMAC), translated into Italian [20]. Furthermore, using the Tinetti scoring system [21,22], we assessed gait and weight bearing during the dynamic phase to verify the patients' ability to reproduce a correct PWB motor pattern. We later interviewed the patients regarding their insole compliance with the NRS, concerning ease of use, absence of complications, and pain with daily use [23].

The standard rehabilitation protocol used in our clinic to treat patients after TKR includes immediate non-weight bearing mobilization with Kinetec (0–60° ROM) and standing upright from the second day post-surgery with partial and progressive weight bearing, starting from 50% of the patients' body weight to 90% at discharge (using a double scale in the static bipodalic position to help patients find the correct bearing). Rehabilitative programs always include foot pumping during the first day, lower limb massage therapy, pharmacologic therapy with 100 mg ketoprofen infusion once per day (to prevent heterotopic ossification), stair climbing, and proprioceptive exercises. For the study group, we requested they stand up from the second day post-op with the insole, and we planned their weight bearing at 30–50% of the body weight. This would then increase according to the following scheme: 40–60% from the 4th day, 50–70% from the 7th day, 60–80% from the 10th day, and 80–90% from the 15th day until discharge (day 30).

The paired *t*-test was used for numerical variables. Null hypotheses of no difference were rejected if two-sided *p*-values were less than 0.05. Data were analyzed statistically using Matlab (MATLAB and Statistics Toolbox Release 2012b, The MathWorks, Inc., Natick, MA, USA).

## 3. Results

No patients needed to quit the study or modify the rehabilitation protocol due to insole-related complications. No patients had post-surgical problems. The clinical outcomes were comparable to the ones obtained from the patients we habitually treat in our orthopedic department with the same surgical technique, surgeon, and rehabilitation program, although with wide interpersonal variability, as shown by the standard deviation (due to the small study cohort of 30 patients versus the 303 control patients).

The patients' mean pre-operative NRS value was $6.71 \pm 1.94$, which remained almost constant after 5 days post-op ($6.08 \pm 1.93$) ($p = 0.301$) but decreased significantly to $4.10 \pm 3.11$ after 30 days ($p = 0.043$). The average post-operative standard KSS score was $53.33 \pm 6.44$, while the functional KSS score was $65.42 \pm 23.11$, a remarkable interpersonal difference ($p = 0.047$). At day 5 post-surgery, the standard KSS score decreased to $49.55 \pm 19.55$ (not significantly, $p = 0.241$), while the functional score reduced significantly to $11.72 \pm 10.71$ ($p = 0.0001$). This was still widely variable in relation to the functional starting conditions at time 0.

At day 10 post-op, the patients' average KSS score was stable ($50.73 \pm 11.71$), while the functional score improved significantly to $45.55 \pm 9.40$ ($p = 0.0032$). After skin suture removal and wound dressing renewal (from advanced to standard dressing) on day 14 post-op, standard KSS score rose to $63.00 \pm 15.03$ ($p = 0.0532$) at day 15 and the functional KSS component to $49.09 \pm 11.36$ ($p = 0.011$), while after a final control on day 30, the discharged patient cohort had means of $78.88 \pm 12.89$ ($p = 0.001$) and $71.42 \pm 15.20$, respectively. The WOMAC scores confirmed a similar pattern and increased from $45.69 \pm 15.96$ pre-op to $51.37 \pm 15.05$ on day 5, then significantly decreased definitively to $14.29 \pm 7.52$ in the final evaluation on day 30 ($p = 0.001$).

The Tinetti gait score showed that, after initial difficulty maintaining a correct deambulation pattern with respect to the recommended load range, all patients reached a normal gait according to direct therapist evaluation and Tinetti test scores. In fact, they started from a pre-operative mean value of $8.08 \pm 2.88$, went to $5.08 \pm 1.88$ on day 5, $6.50 \pm 1.57$ on day 10, and $7.67 \pm 1.37$ on day 15, and then had a definitive value of $9.43 \pm 1.90$ (Table 1).

**Table 1.** Results with different scores: NRS: Numerical Rating Scale; KSS stn: Knee Score System, standard component; KSS FUN: Knee Score System, functional component; WOMAC: Western Ontario McMaster Universities Index Osteoarthritis; Tinetti: Tinetti scoring system.

| TIME (Days Post-Op) | NRS | KSS Stn | KSS FUN | WOMAC | Tinetti |
|---|---|---|---|---|---|
| 0 | $6.71 \pm 1.94$ | $53.33 \pm 6.44$ | $65.42 \pm 23.11$ | $45.69 \pm 15.96$ | $8.08 \pm 2.87$ |
| 5 | $6.08 \pm 1.93$ | $49.55 \pm 19.55$ | $11.73 \pm 10.72$ | $51.37 \pm 15.05$ | $5.08 \pm 1.88$ |
| 10 | $4.58 \pm 1.24$ | $50.73 \pm 11.71$ | $45.55 \pm 9.40$ | $47.18 \pm 15.04$ | $6.50 \pm 1.57$ |
| 15 | $3.50 \pm 1.17$ | $63.00 \pm 15.03$ | $49.09 \pm 11.36$ | $41.83 \pm 17.22$ | $7.67 \pm 1.37$ |
| 30 | $4.10 \pm 3.11$ | $78.88 \pm 12.89$ | $71.43 \pm 15.20$ | $14.29 \pm 7.52$ | $9.43 \pm 1.90$ |

In the first post-operative days, all patients found it impossible to meet the touch-down weight-bearing load, corresponding to a load of 10–20% bodyweight without walking alterations (flexed knee, ankle plantar flexion with forefoot footing) or gait pattern modifications from a continuous walking march to a stepping march (such as little jumps). Only four patients on the third day post-op could walk normally with a load between 30% and 50% of their bodyweight.

Furthermore, we recorded that load beyond 80–90% of bodyweight was not detectable during dynamic gait. Patients behaved as though they were applying full load (i.e., 100% bodyweight). Only one patient did not correctly load because of a neurological quadriceps deficit resulting from a previous lumbar disc herniation.

All patients were able to reach a complete load (90% bodyweight or more) before discharge and showed good adherence to the rehabilitation protocol and insole use. The final insole NRS score, ranging from 0 (poor satisfaction) to 10 (maximal approval), had an average of 8.5, with only one patient complaining of some difficulties in using the device; however, she had an ankle sprain that had occurred just before the surgery.

The number of patients able or unable to achieve the load targets in function of the number of days post-op are shown in Figure 3.

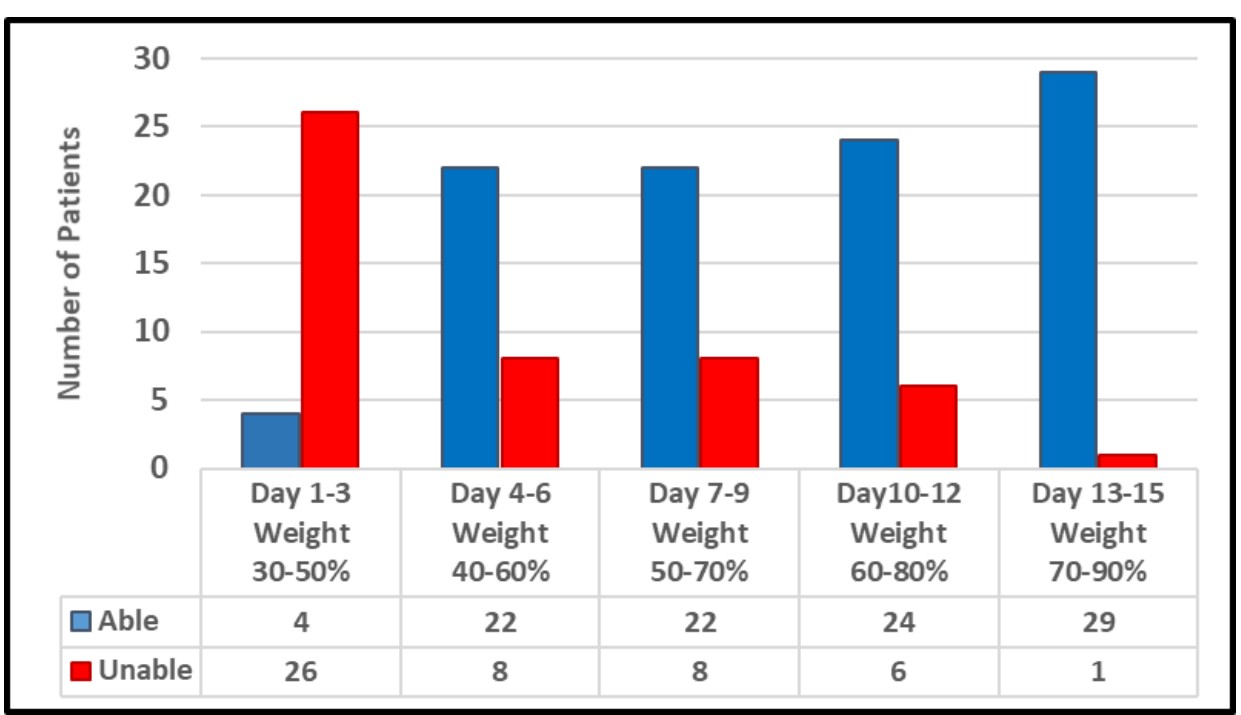

**Figure 3.** Patients able or unable to achieve the weight-bearing targets at various days post-op. All enrolled patients except one succeeded in loading the lower limb with >80% bodyweight at day 15. One other patient could not reach full weight bearing load at day 15 post-op due to a neurological deficit relapse with cruralgia secondary to degenerative lumbar discopathy.

## 4. Discussion

We investigated the use of a smart insole during the rehabilitation of patients who underwent total knee replacement. We found that the use of this device led to several advantages: it improved patients compliance and willingness to follow the rehabilitation protocol, it provided better assessment by the physiotherapist of patients' post-op progress, and it improved patients' overall satisfaction.

Sometimes after discharge from rehabilitation centers, it is difficult to follow up with outpatients in their domestic activities to monitor their gait, load, and functional daily activities [24]. In addition, patients instructed not to weight load on the operated limb sometimes do not load it sufficiently when they are instructed to do so, leading to different complications (e.g., algodystrophy, venous blood return deficit, muscular atrophy, and bone loss) and a consequent delay in complete recovery [25]. Hershko [4] demonstrated that the use of some kind of bio-feedback device improves patients' ability to comply with rehabilitation protocol, therefore making their recovery more efficient and effective. Traditional ways of controlling patient weight bearing during rehabilitation are based on physiotherapists' subjective inputs, and therefore patients cannot be trained in an ideal way. The present study has some limitations as it was a single-arm investigation and the analyzed patient cohort was relatively small. However, our findings seem to support the routine use of the instrumented Blu Insole as a valid and easy-to-use device during rehabilitation after TKA surgery, with the aim of consolidating physiotherapy protocols;

reducing recovery stay; and, above all, improving functional recovery, especially in young, active patients. The rise of patient accountability and deeper involvement during their rehabilitation made them more active and cooperative, with a further improvement in results and compliance.

Our findings seem to support the extension of insole use to all lower limb surgeries that require progressive weight bearing. Without a doubt, it is of high importance to quantify and register this rehabilitation method, making it a provable, reproducible, and comparable parameter in orthopedic and traumatologic lower limb surgeries. It is also possible to use the insole after some months or years to quickly monitor limb maintenance without additional costs. Our findings are also in agreement with the systematic review of Ngueleu et al. [26], who reported strong improvement in rehabilitation with instrumented insole; with the study of Maculewicz et al. [27], that state that even if the main focus was on Parkinson disease, that in order to build a successful systems for patients' gait rehabilitation, the use of technological solutions is beneficial; and with the recent study of Subramanina et al. [28], that states that wearable health monitoring devices allow for measuring physiological parameters without restricting individuals' daily activities, providing information that is reflective of an individual's health and well-being.

We believe we must further extend the application of this device to analogous areas to evaluate focused protocols depending on the surgery (e.g., unicompartmental knee arthroplasty vs. TKA) and the possible use of a double insole to find better solutions, particularly after lower limb arthroplasties (e.g., combined total knee and hip replacements or bilateral TKA).

Further randomized studies with larger patient groups will be needed to confirm our findings.

**Author Contributions:** Conceptualization, G.C. and B.I.; methodology, M.M.; software, M.M., G.B. and R.R.; validation, M.M., G.B. and R.R.; formal analysis, M.M.; investigation, M.M., and E.B.; resources, G.C.; data curation, E.B., and B.I.; writing—original draft preparation, M.M.; writing—review and editing, G.C., M.M., G.B., R.R., E.B., B.I.; visualization, M.M., E.B.; supervision, G.C., B.I.; project administration, G.C., B.I.; funding acquisition, N.A. All authors have read and agreed to the published version of the manuscript.

**Funding:** This research received no external funding.

**Institutional Review Board Statement:** This study was conducted according to the guidelines of the Declaration of Helsinki (7th revision; 64th WMA General Assembly, Brazil, October 2013) and in observance of existing Italian legislation on privacy and sensitive data processing. Prior informed consent was obtained from all subjects involved in the study.

**Informed Consent Statement:** Informed consent was obtained from all subjects involved in the study.

**Data Availability Statement:** The data that support the findings of this study are available on request from the corresponding author.

**Conflicts of Interest:** The authors declare no conflict of interest.

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
