# Peer review of "Effect of an Innovative Biofeedback Insole on Patient Rehabilitation after Total Knee Arthroplasty"

_applsci, doi:10.3390/app12052456_

Round 1

Reviewer 1 Report

Dear Editor,

the manuscript „Effect of an innovative biofeedback insole on patient rehabilitation after total knee arthroplasty” the authors address whether insoles are effective for partial weight-bearing for patients after TKA. However, the methodology shows major weaknesses and the results presented are not sufficiently discussed.

Comments to the author

General comments:

  1. It is claimed that partial weight bearing occurs after any lower limb surgeries. This general statement is not valid. The current clinical evidence shows that full weight-bearing is being sought earlier and earlier after various operations.
  2. Even the concepts of "fast-track" after TKA, which in most concepts provides for full weight-bearing a few hours after the operation.  In this context, the question examined here does not make sense or should be explained in a different contex.
  3. In my eyes,in the methodology is missing a paragraph on the data processing of the insoles and the implementation of a statistical evaluation.
  4. Are the insoles used here validated? If yes, please provide sources. If no, they are not suitable for a scientific study (validity of the results?). The insoles have to be tested against the gold standard a force plate.
  5. The discussion seems clearly too short. There follows no comparison to the current literature or limitations of the study.

Specific comments:

  1. Line 14: “Partial weight bearing is fundamental to rehabilitation following lower limb surgery” see above. I think this general statement is no longer valid today.
  2. Line 72: This should be part of the method section and therefore be “2.1 Patients” instead of “3. Patients”
  3. Line 94: Same as in Line 72; this should be 2.2 Hardware
  4. Line 99: What means “.. up tp 214 FSRs”? The number of used FSRs is important for the validity of the results
  5. Line 112: How is the battery charged? Is this user-friendly, especially for elderly patients?
  6. In this context: How is the recorded data stored? How is the data transferred? How is the data processed? What filters are used in the data processing?
  7. Line 148: …a significant reduction. With a mean of 4.14 and a standard deviation of 3.58, I doubt a statistically meaningful reduction. Why are no p-values given in the whole paragraph? How were the values compared (t-test etc.?)?
  8. Table 1: Here is closed parentheses missing (days Post-Op), furthermore you should provide p-values in the table.
  9. Line 175: “only 4 patients on day 30 could walk normally with a load between 30% and 50%..” So all 26 other patients could not weight bearing 30 days post-op? Or did they more weight? Describe their results in more detail! Use a graphical representation.
  10. Line 178: what means, that load differences beyond 80-90% were not detectable?
  11. Line 188-189: “..Insole as a valid, effective,.. device” Where do you proof this statement?
  12. Line 190: “Reduce recovery stay” and “improve functional recovery” compared to whom or what? You did not have any control group, so you don’t know if the use of Insoles lead to better outcomes.
  13. Line 193-198: You should cite some literature here.
  14. Figure 1: You should describe the Insole more detailed.
  15. Table2 (what should be a figure) Title? X-axis lable? Legend? You also do not reference to this figure in your text.
  16. The table above is doubled in the manuscript (see table1)

Author Response

Our answers to Reviewer #1 together with the amendments made to the text are listed in the document we attached.

Reviewer 2 Report

I have read the article carefully, I see many gaps, and that what they propose is not very reproducible.
 Putting templates to everyone who trades a PTR to see if they carry 30% of the weight or more or less, requires more justification.
I find the study very interesting, since I trust the templates a lot
I understand that it lacks drafting and methodology
The template I think would be better with podiatry correctors and podoscope.

 The methodology:

-In the abstract they talk about the need to establish a correct gait, but what defines a correct gait? p

-In the introduction they say that a load greater than 30% can trigger loosening of the prosthesis, reference 1 which is a study of templates. I think that statement should be supported by something more consistent.

-Line 45: they talk about traditional monitoring methods but do not say what they are and why they consider them obsolete.

-Line 75: they say that it is impossible to make a control group. You can make a control in a group without a template, or put a "placebo" template so that the patient is being measured, but not.

-Line 86: what comorbidities are an exclusion factor? diabetes? Vasculopathy? be operated on the other knee? They are factors that I consider that can affect the result but they do not tell us about them.

-As for the selection of patients, they do not tell us about the degree of deviation of the knee axis, I do not think that a patient with 3 degrees of valgus (physiological) is the same as a patient with 20 degrees of varus, even having the same same degree of osteoarthritis ...

-As for surgery: is the prosthesis cemented in all its components? in any, in none? the patella is fitted. They tell us that in all patients they do the same and the brand of the prosthesis they put on, but nothing more. There are many different techniques that can affect gait even with the same prosthesis. They don't explain it.

-Results:

-Line 189: they say that the device is very easy to use. to explain better

-Line 189: again they say that it improves the postoperative period and recovery. But don't compare it to anything! how can they say that? based on what?

-Line 199: They say that their study supports the use of the template in question in all lower limb surgery that requires a partial load in the postoperative period. Being a study that only talks about PTRs, that the psotoperative of any other surgery of the lower limb may or may not have anything to do with it.

-Line 202: they say that this device is also useful to monitor gait even months or years later. Based on what? if they only study it for 30 days. to explain better

The sample is very small, and the statistics must be inferential, a T or ancova test is necessary
In addition, it is necessary to use other independent variables that can support the article.

Author Response

Our answers to the Reviewer #2 comments together with the text amendments are listed in the attached document

Reviewer 3 Report

This paper requires a spelling check in order to be published. In this sense, a thorough revision of the entire text must be made.
Some of the errors found are shown below:

Patiente Rehabilitation After Total Knee Arthroplasty- patient

programmable biofeedback insole for partial weigh tbearing- weight bearing

Patients secletion was performed according  -selection

BLU Insole allow the (simulatanous?) use of 214 FSR improving -parentheses and questions

Our results support the instrumentes insole- instruments

Author Response

Our answers to Reviwers #3 remarks are listed in the attached document.

Round 2

Reviewer 2 Report

the article may be interesting, since it deals with a new topic, such as insoles, however, it lacks sufficient methodological rigor to be published.

First of all, you need to use the magazine template.

secondly, it is necessary to use tables that accompany the results, right now it is very difficult to interpret.

Finally, it is pertinent to point out the conclusions of the study.

For all this I suggest using and filling in the magazine template and consider later.

Author Response

The manuscript and the tables has been re-edited and adapted to the format required by the Journal.

Here attached, please find all the relevant material.
